# Liver Fetuin-A at Initiation of Insulin Resistance

**DOI:** 10.3390/metabo12111023

**Published:** 2022-10-25

**Authors:** Nicolas Lanthier, Valérie Lebrun, Olivier Molendi-Coste, Nico van Rooijen, Isabelle A. Leclercq

**Affiliations:** 1Laboratory of Gastroenterology and Hepatology, Institut de Recherche Expérimentale et Clinique, UCLouvain, 1200 Brussels, Belgium; 2Service d’Hépato-Gastroentérologie, Cliniques universitaires Saint-Luc, UCLouvain, 1200 Brussels, Belgium; 3Department of Molecular Cell Biology, Vrije Universiteit Medical Center, 1081 Amsterdam, The Netherlands

**Keywords:** fetuin-A, hepatokine, immunofluorescence, nonalcoholic fatty liver disease, fatty liver, steatosis, macrophage, insulin resistance, prediabetes, Kupffer cell

## Abstract

Hepatokines (liver secreted proteins with possible distant action) are emerging potential players in insulin resistance in type 2 diabetic patients. Here, we explored the effect of a high-fat diet on the expression of fetuin-A, one of those candidate liver proteins, and its relationship with liver macrophage activation. Mice were fed a normal diet or a high-fat diet for 3 days, known to initiate steatosis and liver insulin resistance. A preventive liver macrophage depletion was obtained by intravenous injection of clodronate-loaded liposomes. The mRNA and protein expression of fetuin-A was evaluated by qPCR, Western blot and immunofluorescence on different insulin-sensitive tissues (liver, adipose tissue, and muscle). Short-term high-fat diet-induced steatosis, liver macrophage activation, and hepatic insulin resistance together with a significantly increased expression of liver AHSG (α2-HS glycoprotein/fetuin-A) mRNA and serum fetuin-A concentration. On immunofluorescence, fetuin-A was mostly expressed in centrilobular hepatocytes. This increase in fetuin-A under high-fat diet was not evidenced in other peripheral insulin-sensitive tissues (skeletal muscle and adipose tissue). The mRNA expression of α2-HS glycoprotein was 800 times higher within the liver compared with the adipose tissue or the muscle. Liver macrophage depletion that significantly ameliorated insulin sensitivity was associated with a significant decrease in α2-HS glycoprotein mRNA expression. In conclusion, this study demonstrated liver fetuin-A overexpression at the initiation of high-fat diet feeding, concurrent with hepatic steatosis and insulin resistance. Targeting liver macrophages in this setting reduced liver α2-HS glycoprotein expression suggesting that fetuin-A acts as an hepatokine with proinsulin resistance effects.

## 1. Introduction

Metabolic dysfunction-associated fatty liver disease (MAFLD) occupies, today, a large place in the spectrum of chronic liver diseases, as it is now the first reason of visit at the hepatology clinic and one of the two main causes of liver transplantation [1,2,3,4,5]. In parallel, a similar increasing prevalence of insulin resistance (IR) which can lead to type 2 diabetes is also observed [6].

Those two entities (MAFLD and IR) are, in fact, closely related [7,8]. Indeed, numerous data show a causal relationship between hepatic steatosis and IR both in animal models [9,10] and in patients with MAFLD [6,11,12,13,14,15]. We and others have evidenced that a short-term high-fat feeding (3 days) in animals induced hepatic steatosis as well as rapid hepatic IR [9,10]. This is concurrent with activation of the innate immune system (liver macrophages) [10,16]. Interestingly, depletion of liver macrophages significantly ameliorated hepatic insulin sensitivity [10,17]. A deleterious activation of the innate immune system specifically in the liver is, thus, playing a pivotal role in hepatic IR in response to high-fat diet (HFD) [10,17]. Liver macrophage expansion is also the first difference seen in liver biopsies of patients with steatosis compared to control patients [18]. In healthy individuals, a short-term high-fat overfeeding (5 days) also induces liver-specific insulin resistance (proven in clamp experiments) with a 26% increase of fasting hepatic glucose production, associated with increased transaminase levels and without any change in body composition or peripheral insulin resistance [19].

Whether the liver (and liver macrophages) contribute to the onset of whole body IR in distant organs such as skeletal muscles and white adipose tissue remains a matter of debate [20] even though there is strong evidence to support it [13,21,22]. The liver plays a major role in amino acid and protein metabolism and provides most of the circulating proteins present in the blood. Steatosis or liver injury could alter the liver proteic secretome with potential (pathological) consequences on distant organs [23,24,25]. Fetuin-A, a liver-secreted protein, is indeed described as implicated in adipose tissue IR [26]. This glycoprotein can be secreted by the steatotic liver following the influx of free fatty acids (FFA), circulated in the blood in high concentration, and act as an endogenous ligand of the toll-like receptor 4 (TLR4) present on the adipocytes [26]. The blocking or administration of fetuin-A, therefore, makes it possible to reduce or worsen, respectively, the IR of the animals [26,27,28]. By analogy with adipose tissue proteins called “adipokines” and described as playing a key role in systemic IR in the context of obesity-associated adiposity (such as tumor necrosis factor-alpha or plasminogen activator inhibitor-1), liver proteins such as fetuin-A are named “hepatokines” [29,30]. We have recently detected increased circulating fetuin-A in foz^-/-^ mice fed an HFD and in patients with MAFLD, associated with IR [31]. We have shown, in nonalcoholic steatohepatitis (NASH, the severe form of the disease), the presence of fetuin-A within activated liver macrophages forming crown-like structures or lipogranulomas [31]. However, we know that IR may appear earlier on an HFD and involve liver-resident macrophages (Kupffer cells) before lipogranuloma formation and the development of NASH [10].

We, therefore, decided to characterize a short-term HFD model with IR that we previously described [10] by evaluating fetuin-A levels in the blood, the liver, the muscle, and the white adipose tissue, three key insulin-sensitive organs involved in IR pathogenesis. We paid specific attention to the connections with liver macrophages, known factors in the onset of hepatic IR [10,17,32].

## 2. Materials and Methods

### 2.1. Animals, Diets, and Treatment

After 1 week acclimatizing, 5 weeks old male C57BL/6J mice were fed *ad libitum* for 3 days the HFD in which 60% of calories are derived from fat (D12492 from Research Diets, New Brunswick, NJ, USA) or the normal diet (ND) (10% of calories from fat, Carfil Quality, Oud-Turnhout, Belgium). The day of culling, the liver, epididymal white adipose tissue, and right quadriceps were rapidly dissected. Portions of tissue were immersed in formalin 4%; the remaining tissue was snap frozen in liquid nitrogen and kept at −80 °C until analyses [33]. Intravenous liposome-encapsulated clodronate was used as previously described to selectively deplete liver macrophages [10,34]. Analyses were performed on 8 animals per group (24 animals in total).

### 2.2. Hyperinsulinemic-Euglycemic Clamp Study

In another set of 18 animals, an intravenous catheter was implanted, and insulin sensitivity was assessed by the euglycemic-hyperinsulinemic clamp as described [10,35,36]. Briefly, 5 h fasted mice were infused with insulin at a rate of 2.5 mU·kg^−1^·min^−1^ for 2.5 h. Glucose was infused at a variable flow rate to maintain euglycemia. For glucose turnover measurements, [3-^3^H] glucose (Perkin Elmer, Boston, MA, USA) was infused at a rate of 0.33 μCi·min^−1^. Analyses were performed on 6 animals per group.

### 2.3. Biochemical Analyses

Total liver lipids were extracted with methanol and chloroform and quantified by the vanillin-phosphoric acid reaction [37]. Serum levels of fetuin-A were measured by a mouse fetuin-A enzyme linked immunosorbent assay (ELISA) kit (R&D systems) [31].

### 2.4. Immunofluorescence

Detection of F4/80 and fetuin-A was performed on formalin-fixed, paraffin-embedded sections treated with proteinase K using a primary rat antimouse F4/80 monoclonal Ab (1/200, Serotec, Oxford, UK) and a primary goat antimouse/human fetuin-A (1/100, Santa Cruz, CA, USA). Secondary antibodies were donkey antirat/AlexaFluor 594 and donkey antigoat/AlexaFluor 488 (dilution 1/2000, Invitrogen, Merelbeke, Belgium). Hoechst (dilution 1/10,000) was used to reveal the nuclei.

### 2.5. Protein Studies–Western Blotting

Proteins from liver, white epidydimal adipose tissue, and muscle homogenates were analyzed by Western blotting using a goat fetuin-A antimouse/human antibody (1/2000, Santa Cruz, CA, USA). The immunoreactivity was detected with a horseradish peroxidase-conjugated secondary antigoat Ab (1/40,000) and enhanced chemiluminescence reagents (Western Lightning Chemiluminescence Reagent Plus, Perkin-Elmer, Boston, MA, USA). Ponceau-S red staining was used as a loading control. One membrane was sequentially probed with the fetuin-A antibody and then with the HSP-90, beta-actin, glyceraldehyde 3-phosphate dehydrogenase (GAPDH) or Akt antibodies to control for protein loading.

### 2.6. RNA Extraction, Reverse Transcription, and RT-qPCR

Total RNA was extracted from frozen liver, epididymal fat, and muscle samples using TRIpure Isolation Reagent (Roche Diagnostics Belgium, Vilvoorde). cDNA was synthesized. Quantitative real-time PCR analysis was carried out as previously described [33]. Primer pairs for transcripts of interest F4/80, CD68, α2-HS glycoprotein (AHSG/fetuin-A), TNF-α, CD68, and RPL-19 chosen as an invariant standard were designed using the Primer Express design software (Applied Biosystems, Lennik, Belgium). Primer sequence for AHSG was 5′-TGGCCTGCAAGTTATTCCAAA-3′ (forward) and 5′-GCTGTGGGTACGGGACCTACT-3′ (reverse) [31]. The sequences of the other primers (F4/80, TNF-α, CD68, and RPL-19) have already been described [10]. All experimental tissues and standard curve samples were run in duplicate in a 96-well reaction plate (MicroAmp Optical, Applied Biosystems). Results are expressed as fold expression relative to expression in the control group using the ΔΔCt method.

### 2.7. Statistical Analysis

All the data are presented as means ± SD. Statistical analysis was performed by using GraphPad Prism^®^ for Windows (v.6.01, La Jolla, CA, USA). All data were checked for normality using the Shapiro–Wilk normality test. Data were analyzed using a one-way ANOVA with Tukey’s post hoc tests for multiple comparisons. Data determined to be non-normal were analyzed using a Kruskal–Wallis test with Dunn’s multiple comparisons post-tests. We considered *p* ≤ 0.05 to be statistically significant.

## 3. Results

### 3.1. Upregulation of AHSG mRNA Expression in the Liver under Short-Term High-Fat Feeding

As previously described [10], short-term HFD mice gained weight (Figure 1A) and developed liver steatosis (Figure 1B) and hepatic IR proven at the clamp study (Figure 1C,D). We measured by RT-qPCR the expression of fetuin-A, a potential hepatokine suspected to contribute to peripheral IR pathogenesis. The liver mRNA level of AHSG was significantly increased in mice fed a 3-day HFD compared to controls fed the ND (Figure 1E). However, the protein content of fetuin-A within the liver did not change (Figure 1I). In accordance with previous findings, we confirmed the activation of resident liver macrophages, evidenced by a significant upregulation of liver F4/80, a cell surface glycoprotein of Kupffer cells (Figure 1F), and an upregulation of liver CD68, a transmembrane glycoprotein of macrophages associated with lysosomal compartment (Figure 1G).

### 3.2. Deletion of Hepatic Macrophages Modulates AHSG Expression

Knowing the roles of both liver macrophages and fetuin-A on insulin sensitivity, we then wanted to explore the effect of liver macrophage depletion on fetuin-A protein expression. This was achieved by an intravenous injection of liposome encapsulated clodronate one day prior to starting the mice on the HFD. As previously shown, intravenous clodronate liposome injection selectively depleted liver macrophage [34]. Indeed, clodronate treatment eliminated liver F4/80 and CD68 transcripts (Figure 1F,G). Consistent with the removal of macrophages, hepatic tumor necrosis factor-alpha (TNF-α) was significantly downregulated in clodronate-treated animals (Figure 1H). We then measured the mRNA level of AHSG in liver-macrophage-depleted animals compared with other groups. Liver macrophage depletion was associated with a significant downregulation of the liver AHSG mRNA level, to a level comparable to that of animals on a control diet (Figure 1E).

### 3.3. Fetuin-A Circulating Form and Distribution within the Liver

High AHSG transcript levels in the liver of short-term HFD fed mice coincided with a significantly higher amount of the circulating fetuin-A protein (Figure 1J). However, while macrophage depletion significantly downregulated AHSG mRNA expression within the liver, it did not lower serum levels (Figure 1J).

On liver histological sections, fetuin-A was mainly located in the centrilobular hepatocytes, showing the granular appearance of secretory vesicles (Figure 2A). The resident macrophages of the liver positive for the F4/80 glycoprotein of the surface (Kupffer cells) were mainly located in the sinusoids adjacent to hepatocytes from the intermediate zone (Figure 2B). Costaining of fetuin-A with the F4/80 antibody did not reveal any colocalization between fetuin-A and liver macrophages (Figure 2C). The same situation was evidenced after the short-term HFD (Figure 2F–H).

### 3.4. Fetuin-A in Other Insulin-Sensitive Tissues

We then compared the fetuin-A levels between the liver and the other two main insulin-sensitive tissues: the white adipose tissue and the skeletal muscle. We loaded the same amount of total protein on a gel for Western blot analysis (100 µg). Ponceau-S red staining confirmed equal protein loading amongst samples and tissues (not shown). The amount of beta-actin or glyceraldehyde 3-phosphate dehydrogenase (GAPDH), commonly used for intersample normalization, greatly varied according to the nature of the tissue (Figure 3A). By contrast, Akt showed a more stable expression across tissues (Figure 3A). Although Akt is part of the insulin signaling pathway, the total form is not affected by the experimental conditions (HFD or clodronate administration) [10] and can, therefore, serve as a reference. The highest amount of fetuin-A was observed in the adipose tissue compared with the liver or the muscle (Figure 3A). However, as in the liver (Figure 1I), HFD was not associated with an increased amount of fetuin-A in the adipose tissue or in the muscle (Figure 3B).

Finally, we compared the mRNA levels of AHSG within those three tissues. The ribosomal protein L19 (RPL19) showed minimal variation in expression across the samples and tissues and was, thus, chosen as the housekeeper gene (Figure 3C). In sharp contrast with the Western blot data, the mRNA expression of AHSG was 800 times higher within the liver compared with the adipose tissue (Figure 3D). The muscle mRNA expression of AHSG was also very low compared with the liver (Figure 3D).

## 4. Discussion

In this study, we characterized fetuin-A expression and distribution in the liver, the blood, the adipose tissue, and the muscle in a short-term HFD model of IR.

We showed that the circulating levels and the liver mRNA expression of AHSG (fetuin-A) are increased after three days of high-fat feeding in mice. This occurred in parallel with the development of liver steatosis, hepatic IR, and liver macrophage activation and underlines the central role of the steatotic liver in the onset of IR. This confirmed a concept previously inferred by the analysis of a large cohort [38]; namely, that fetuin-A is not only a marker of diabetes, IR [39,40], or MAFLD [41,42] but also a trigger for diabetes development [38]. Indeed, in their large prospective study in which nondiabetic women were included, the authors found a positive association between baseline plasma fetuin-A levels and the occurrence of type 2 diabetes during follow-up [38]. A recent interventional study in humans pointed in the same direction [43], showing that a short-term HFD, already known to induce liver steatosis [44] and hepatic IR [19], also induced a low but significant increase in plasma fetuin-A level and is associated with reduced whole body insulin sensitivity [43]. In a large scale population-based study, fetuin-A levels correlated with the fatty liver index (FLI, a noninvasive test based on waist circumference, body mass index, level of triglycerides, and γ-glutamyl transpeptidase) indicating hepatic steatosis [45]. Importantly, FLI predicts the risk of type 2 diabetes development in people with prediabetes [15]. Our results on the rapid increase of both liver AHSG mRNA levels and fetuin-A serum levels reinforced our belief that the steatotic liver plays a central role in the pathogenesis of IR and supported the concept that this phenomenon could be mediated by fetuin-A.

Second, our results highlighted that fetuin-A, at least in part, is produced by the liver. Indeed, the high mRNA content of fetuin-A within the liver (compared with levels in other insulin-sensitive tissues) and the strong immunofluorescence signals in centrilobular hepatocytes are consistent with the liver being a major site for fetuin-A production. Such detection of fetuin-A by immunofluorescence in wild-type mice demonstrated fetuin-A production by hepatocytes and its accumulation in granular structures consistent with Golgi apparatus or secretory granules. The detection of fetuin-A in the liver has already been attempted in the past [46]. We recently demonstrated its histological pattern in foz^-/-^ mice and in patients with MAFLD showing the same results in the case of isolated steatosis, before the development of NASH [31]. Although we did not provide evidence that these are secretory vesicles, this picture of a granular formation resembles that of other secreted proteins such as albumin [47].

Third, our results showed that the hepatic expression of AHSG is modulated by hepatic inflammation. A short exposure to HFD induced both liver inflammation and fetuin-A production. Moreover, blunting HFD-induced inflammation by depleting liver macrophages decreased AHSG mRNA levels to a level found in ND-fed animals. Together with the immunofluorescent data, it confirms that fetuin-A is produced by hepatocytes (not by macrophages) and that its production is mediated by the inflammatory milieu. Despite this, high fetuin-A serum concentration does not drop upon macrophage depletion. The reason for this remains unknown. This might be in link with a relatively long half-life of the protein in the circulation and our very short experimental design. One alternative explanation is that the clearance mechanisms of fetuin-A involve liver macrophages. Concerning the discrepancy between the high level of the protein in the blood and the stable level of the protein in the liver (in immunofluorescence and Western blot), we imagine that it is related to the fact that the protein is rapidly secreted by the liver. This explains why we observed an increase in hepatic AHSG mRNA on a high-fat diet, a stable level of liver fetuin-A protein, and an increase in the circulating form in the blood. This increase in circulating protein correlating with the increase in hepatic messenger RNA has been described for other secreted hepatic proteins such as selenoprotein-P [48].

The high fetuin-A protein presence in the adipose tissue with a low AHSG mRNA level is consistent with a trapping of circulating fetuin-A in the adipose tissue, rather than with a local production. Cell culture experiments showing fetuin-A in the supernatant of cultured hepatocytes but not in cultured adipocytes support this hypothesis [31,49]. The study of humans with obesity offered similar data with a high fetuin-A protein level measured in the adipose tissue while its mRNA was undetectable [49]. Collectively, our results supported the concept that fetuin-A potentially acts as a hepatokine stimulated by high-fat feeding, secreted into the bloodstream, and that it accumulates in peripheral sinks such as the adipose tissue [31]. These data provided insight into the early mechanisms of development of IR following initiation of an HFD and confirm the central role of the steatotic liver in the pathogenesis of complications. Hepatic macrophages and the liver secretome should be considered both as markers of IR and as potential therapeutic targets in MAFLD [32,41,50,51,52,53]. Tissue selective depletion of liver fetuin-A will support our findings and provide evidence for a causal role of liver fetuin-A production in insulin resistance and accumulation of this protein in adipose tissue.

In conclusion, the present study demonstrated that hepatic mRNA levels and circulating fetuin-A levels are elevated in response to a short-term high-fat feeding in association with steatosis and hepatic insulin resistance. Our findings also showed that this fetuin-A production is localized at the hepatocyte level and its expression is modulated by liver macrophages.

## Figures and Tables

**Figure 1 metabolites-12-01023-f001:**
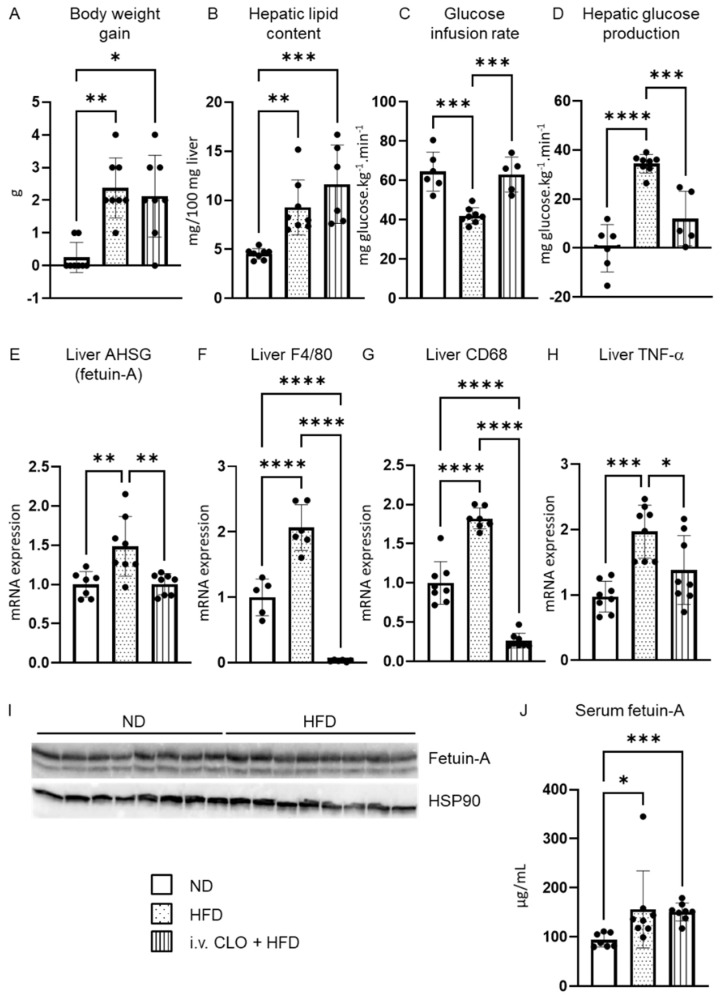
Liver fetuin-A is increased upon short-term high-fat diet and is modulated by liver macrophages. Body weight (**A**), hepatic lipid content (**B**), glucose infusion rate (**C**), and hepatic glucose production (HGP) (**D**) changes in normal diet (ND) mice, 3-day high-fat diet (HFD) mice and 3-day HFD mice with liver macrophage depletion obtained by intravenous (i.v.) clodronate (CLO) injection. Liver mRNA expression of AHSG (**E**), F4/80 (**F**), CD68 (**G**), and TNF-α (**H**). Liver fetuin-A protein (**I**) and serum fetuin-A (**J**) protein expression. Data are expressed as means ± SD. *n* = 6–8 per group. * *p* < 0.05, ** *p* < 0.01, *** *p* < 0.001, **** *p* < 0.0001.

**Figure 2 metabolites-12-01023-f002:**
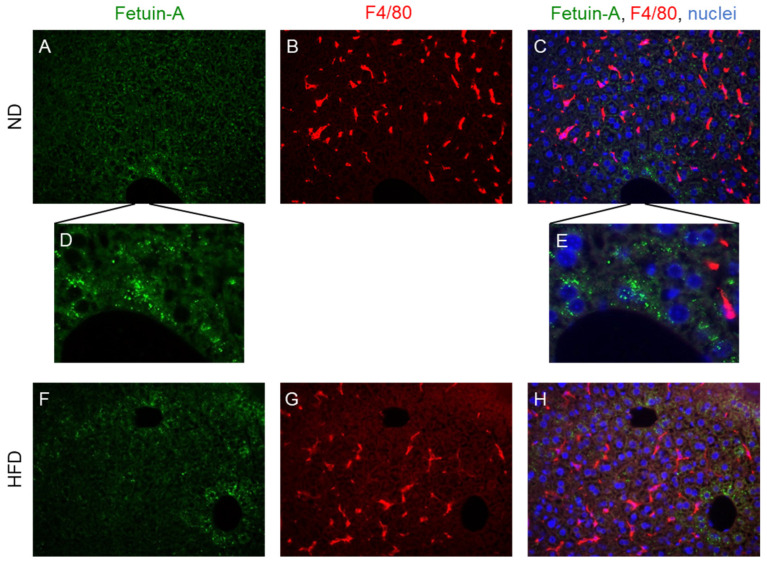
Fetuin-A distribution in the liver. Representative immunofluorescence detection of fetuin-A (green) (**A**,**C**,**D**–**F**,**H**), F4/80 positive cells (red) (**B**,**C**,**E**,**G**,**H**) and nuclei (blue) (**C**,**E**,**F**) in liver sections of a normal diet (ND)-fed animal (**A**–**E**) and a high-fat diet (HFD)-fed animal (**F**–**H**). The magnification (**D**,**E**) allows the visualization of the granular aspect of fetuin-A within the hepatocytes.

**Figure 3 metabolites-12-01023-f003:**
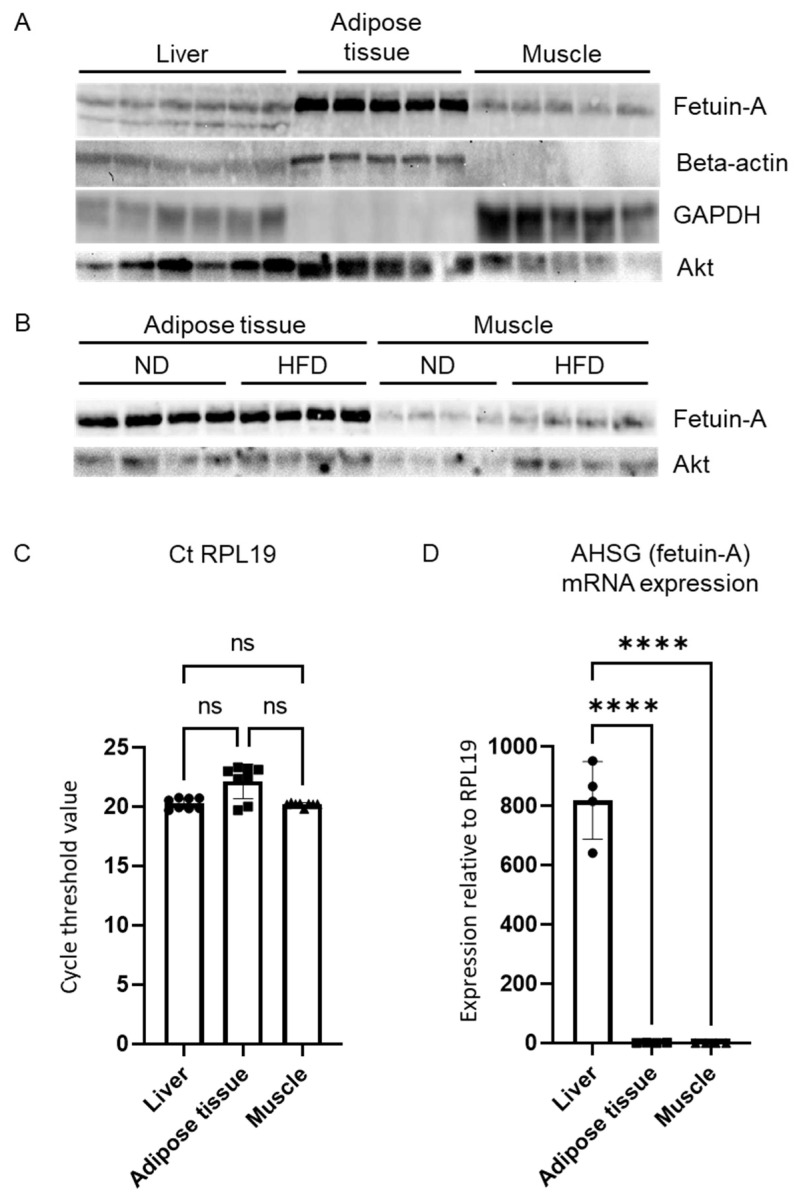
Fetuin-A in insulin sensitive tissues. Comparison of liver, adipose tissue, and muscle protein content of fetuin-A, beta-actin, glyceraldehyde 3-phosphate dehydrogenase (GAPDH), and Akt evaluated by Western blot analysis in mice fed the ND (**A**,**B**) or the HFD for 3 days (**B**). Mean cycle threshold values of RPL19 in the liver, adipose tissue, and muscle (**C**) evaluated by qPCR analysis. Comparison of liver, adipose tissue, and muscle AHSG mRNA expression with RPL19 mRNA chosen as an invariant standard (**D**). Data are expressed as means ± SD. *n* = 3–6 per group. ns: nonsignificant, **** *p* < 0.0001.

## Data Availability

All data analysed during this study are included in this published article. The datasets generated during and/or analysed during the current study are available from the corresponding author on reasonable request.

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
