# Peer review of "Liver Fetuin-A at Initiation of Insulin Resistance"

_metabolites, 2022, doi:10.3390/metabo12111023_

Round 1

Reviewer 1 Report

In the study “Liver fetuin-A at initiation of insulin resistance” the authors explore the effects of a high fat diet on the expression of the hepatokine fetuin-A, and its relationship with the activation of liver macrophages. They found that a short-term high fat diet induced steatosis, liver macrophage activation and hepatic insulin resistance, along with a significant increase in liver fetuin-A mRNA level and serum fetuin-A concentration. This mRNA increase was not detected in other insulin-sensitive tissues, thus suggesting that the liver is the primary source of fetuinA. The effect is mediated by hepatic macrophages activation because their depletion significantly reduced fetuin-A mRNA expression and improve insulin sensitivity. These findings correlate increased fetuin-A expression with hepatic steatosis and insulin resistance and suggest that hepatic macrophages (and the liver secretome) should be considered potential therapeutic targets in MAFLD and diabetes.

The topic is of interest although the association of fetuinA and insulin resistance is already known (htts://doi.org/10.1016/j.bbrc.2006.09.071) as well as the association of fetuinA and macrophage activation (a similar paper has been recently published by the authors on Metabolites, ref 36). The authors should better clarify the novelty of their findings in the text. Furthermore, if the authors’ intent is to emphasize that liver FetuinA is involved in the phenomenon, tissue specific fetuin-A null-mice should be produced and characterized. In the current form, the experiments mainly correlate fetuinA levels with insulin resistance. Additional experiments must be performed to demonstrate the real fetuinA involvement in the phenomenon.

Fig 2. Based on representative images provided, it is difficult to appreciate the expression of fetuinA in centrilobular hepatocytes as well as its granular appearance in secretory vesicles. High magnification images and colocalization experiments with markers of secretory vesicles must be provided.

Fig 3. Several housekeeping proteins were used (GADPH, beta actin) in figure 3A but only Akt selected as a marker of equal protein loading in fig3B. Given the involvement of the Akt pathway in insulin resistance, it is not probably the appropriate housekeeping marker. Authors should test HSP90.

The authors’ intent is to correlate FetuinA levels with insulin resistance, given the relevance of the Akt pathway in insulin signalling, the P-Akt levels should be measured under different experimental conditions.

HFD caused increased fetuinA mRNA levels in liver and protein concentration in serum, without altering its total protein expression in the liver. Conversely, macrophages depletion prevents the increase of fetuinA mRNA and protein levels in the liver, without altering their circulating concentration. The authors should better explain the apparent discrepancy in fetuinA protein content between liver and circulation, under different experimental conditions.

The authors suggest that a trapping phenomenon is responsible for the high fetuin-A protein levels found in the adipose tissue. Measurements of adipose tissue/muscle fetuinA protein expression under macrophages depleted conditions, should confirm the hypothesis

Reviewer 2 Report

The authors in this manuscript want to investigate the effect of a high fat diet on the expression of fetuin-A, a liver protein, and its relationship with liver macrophage activation using mice as an animal model. The mRNA and protein expression of fetuin-A was evaluated by quantitative real-time PCR, Western-blot and immunofluorescence on different insulin-sensitive tissues.

Major points:

In the paragraph “RNA extraction, reverse transcription and RT-qPCR” of materials and methods section it is necessary to describe the conditions used for the quantitative real-time PCR and the primers that were used and how the data were analyzed.

The primers described in lines 119 and 120 must be written as follows: 

5’-TGGCCTGCAAGTTATTCCAAA-3’ (forward) and 5’-GCTGTGGGTACGGGACCTACT-3’ (reverse)

Reference 35 does not report the primers or the PCR conditions for these primers.

Reference 10 does not report primers or PCR conditions for the genes F4/80, TNF-, CD68 and RPL-19.

The abbreviation for quantitative real-time PCR is “qPCR”.

In paragraphs 2.1 and 2.2 the groups of animals must be described, and the total number of animals used must be specified.

When referring to the transcript for the fetuin-A protein the name and symbol GeneCards should be used “AHSG (Alpha 2-HS Glycoprotein)”.

Minor points

In the abstract:

Line 17 replace “RT-PCR” with “qPCR”.

Line 27 replace “HFD” with “high fat diet (HFD)”.

In Materials and Methods:

Line 76 “ad libitum” will be written in Italic.

Legend of Figure 3:

Line 204 replace “real-time PCR analysis” with “qPCR analysis”

References:

Reference number 1 is missing.

Round 2

Reviewer 1 Report

Thanks for the reply. I understand the problem in finding a unique marker, however, I emphasize that the blot staining with Akt in figure 3A is not very convincing.

On line 205 the authors state, "In contrast, Akt showed more stable expression across the tissues (Figure 3A)." If I look at the blot, honestly the Akt signal is barely detectable in the adipose tissue and invisible in the muscle. Paradoxically, the staining is more convincing in the whole blot shown as supplementary material.

So, either replace the Akt line in the blot of figure 3A (probably by increasing the amplitude of the strip, you could better grasp the sign of the protein) or show the Ponceau-S red staining of the blot (eventually also as a supplementary material).

Reviewer 2 Report

The authors have responded adequately to my requests and therefore I believe that the manuscript can be accepted for publication.

Author Response

Thank you for your comment.